# The Use of Sternum and Sacrum Angles in the Assessment of Sitting Posture in Adolescents: A Cross-Sectional Comparison of Cohorts Assessed Before and After the COVID-19 Pandemic

**DOI:** 10.3390/children12111547

**Published:** 2025-11-15

**Authors:** Sun-Young Ha, Arkadiusz Żurawski, Wojciech Kiebzak

**Affiliations:** 1Department of Physical Therapy, College of Health Sciences, Kyungnam University, Changwon 51767, Republic of Korea; mallows205@naver.com; 2Collegium Medicum, Jan Kochanowski University in Kielce, al. IX Wieków Kielc 19a, 25-516 Kielce, Poland; wojciech.kiebzak@ujk.edu.pl

**Keywords:** alignment, sternal angle, sacral angle, COVID-19, kyphosis, lordosis

## Abstract

**Highlights:**

**What are the main findings?**
In two cross-sectional cohorts assessed before and after the COVID-19 pandemic, standardized sitting spinal alignment parameters showed no large between-group differences.The sternal angle correlated with thoracic kyphosis, and the sacral angle with lumbar lordosis, indicating their consistent geometric association within the postural chain.

**What are the implications of the main findings?**
A sternal angle of approximately 65° was consistently associated with thoracic and lumbar curvatures considered physiologically favorable; this may serve as a practical reference value rather than a prescriptive “optimal” posture.Angle-based assessment offers a simple, measurable approach that can support posture education and screening in schools without the need for specialized equipment.

**Abstract:**

**Background**: The COVID-19 pandemic has been associated with increased sedentary behavior in children, raising concerns about posture and spinal health. This study compared standardized measures of sitting spinal alignment in two independent cohorts assessed before (2017) and after (2024) the pandemic and examined correlations among alignment parameters across different sitting postures. **Methods**: This cross-sectional study included healthy children aged 9–13 years. The sternal angle, sacral angle, lumbar lordosis, thoracic kyphosis, trunk tilt, and lateral deviation were measured using a Saunders digital inclinometer and a DIERS Formetric 4D rasterstereographic system in passive, forced, and corrected sitting postures. **Results**: No statistically significant differences were observed between the 2017 and 2024 cohorts (*p* > 0.05). Within each cohort, significant posture-related differences were found for the sternal and sacral angles, lumbar lordosis, and lateral deviation (*p* < 0.05), while thoracic kyphosis and trunk inclination differed between passive and corrected sitting (*p* < 0.05). The sternal angle correlated moderately to strongly with thoracic kyphosis (*r* = 0.657–0.695, *p* < 0.001), and the sacral angle correlated with lumbar lordosis (*r* = 0.679–0.743, *p* < 0.001). **Conclusions**: Similar alignment parameters across time-separated cohorts suggest no major cohort-level shifts in standardized sitting posture; however, behavioral factors were not directly assessed. Strong correlations among sagittal angles emphasize the consistent geometric relationship between the sternum, sacrum, and spinal curvatures. A sternal angle of approximately 65° was consistently associated with physiologically favorable spinal alignment and may serve as a practical reference value for posture assessment and education.

## 1. Introduction

The COVID-19 pandemic, caused by the SARS-CoV-2 virus, led to worldwide lockdowns and prolonged school closures, profoundly changing children’s daily activity patterns [1,2]. In Poland, remote education was implemented in March 2020, forcing millions of students to spend long hours studying online [3,4]. The transition to online learning has been associated with decreased physical activity, reduced outdoor exposure, and increased sedentary time-factors known to negatively affect musculoskeletal development [5,6,7,8,9]. Several studies reported a rise in postural deviations among school-age children during this period, including increased forward head posture, shoulder protraction, and spinal deformities such as scoliosis [10,11,12].

Sitting is one of the most frequent daily positions, and its duration has further increased during recent years due to digital learning and leisure screen use. Poor sitting posture-characterized by a flexed spine, posterior pelvic tilt, and forward head and shoulder displacement-has been identified as a major contributor to spinal misalignment in youth [13,14,15]. Although verbal instructions are often used to correct these deviations, their effectiveness is limited because the target position is difficult to define and verify [16,17]. Quantitative assessment of spinal alignment and the development of simple reference cues are therefore crucial for posture education and screening.

Previous studies have highlighted the importance of the sternum and sacrum angles in describing sagittal alignment and pelvic orientation [18,19]. Kiebzak et al. (2022) demonstrated that simultaneous alignment of the sternum and sacrum serves as a reliable external marker of sitting posture, with a sternal angle of approximately 64° corresponding to physiological thoracic kyphosis and lumbar lordosis in children [18]. Building upon these findings, Kiebzak et al. (2024) showed that forced straightening of the back-an action commonly associated with the verbal instruction “sit up straight”-can distort natural spinal curvatures and lead to an unbalanced, mechanically unfavorable posture [20]. These results underline the clinical need to distinguish between various sitting strategies and to identify geometrically optimal postural cues.

Accordingly, the present study analyzed three standardized sitting conditions-passive, forced, and corrected-to reflect different biomechanical and behavioral postural patterns. The passive posture represents the habitual relaxed “slumped” position typical in daily life; the forced posture corresponds to the excessive straightening induced by verbal instruction and scapular retraction; and the corrected posture, based on clinical principles introduced by Kiebzak et al. [18,20], involves an anterior pelvic tilt and a lifted sternum, resulting in a more balanced sagittal alignment. Considering these distinct characteristics, the inclusion of all three postures allows a comprehensive evaluation of both spontaneous and instructed behaviors, as well as an optimized, clinically guided corrective strategy.

Therefore, the present study aimed to answer the following research question: Do standardized measures of sitting spinal alignment differ between cohorts assessed before and after the COVID-19 pandemic, and how are the sternum, sacrum, and spinal curvatures geometrically related across passive, forced, and corrected sitting postures?

We hypothesized that (1) no major between-cohort differences would be observed under standardized laboratory conditions, and (2) the corrected posture would demonstrate the most physiologically favorable geometric relationships among sagittal alignment parameters.

## 2. Materials and Methods

### 2.1. Participants

This was a cross-sectional study involving two independent cohorts assessed in 2017 and 2024. Participants were children aged 9–13 years who attended the Children’s Hospital in Kielce, Poland, for routine postural evaluations.

The inclusion criteria were:A score of 0 or 1 on the WHO ECOG scale, indicating normal systemic health;Normal thoracic and spinal structures as verified by an experienced physical therapist.

The exclusion criteria included:Spinal pain within the past 3 months;Difficulty identifying the sacrococcygeal joint due to deformities of the sacrum or coccyx;Neuromuscular disorders affecting posture correction or maintenance;Participation in specialized postural disorder therapy;Clinically diagnosed scoliosis;History of spinal or chest surgery, analgesic treatment affecting posture, or deformities of the sternum or thoracic cage.

The study protocol was approved by the Ethics Committee of the Faculty of Medicine and Health Sciences, Jan Kochanowski University in Kielce (Approval No. 17/2016), and conducted in accordance with the Declaration of Helsinki. Written informed consent was obtained from all participants and their legal guardians. Verbal assent was also obtained from the children after detailed explanation of the procedures.

The general characteristics of both cohorts are presented in Table 1.

### 2.2. Procedure

All participants sat barefoot, with their knees and hips flexed 90°, feet flat on the floor, and arms resting loosely at their sides. The examiner instructed participants to look straight ahead to minimize head movement and maintain a consistent trunk posture. No backrests or external supports were used to eliminate compensatory stabilization. Participants were placed in three postural conditions (passive sitting, forced sitting, and corrected sitting), during which the examiner assessed their posture and spinal alignment.

The three postural conditions were as follows:(1)Passive sitting—a habitual posture without instruction or correction.(2)Forced sitting—maximal voluntary extension posture based on verbal instruction from the examiner.(3)Corrected sitting—corrected posture to optimize alignment based on the examiner’s handling.

The DIERS Formetric 4D rasterstereographic system and the Saunders digital inclinometer were used to assess spinal alignment.

### 2.3. Measures

#### 2.3.1. Saunders Digital Inclinometer

The sternal and sacral angles were measured in the sagittal plane using a Saunders digital inclinometer (Baseline Digital Inclinometer, The Saunders Group Inc., Chaska, MN, USA) once in each of three postures [19]. The sternal angle was measured by placing the inclinometer over the sternum in the sagittal plane, and the sacral angle was measured by placing the inclinometer over the mid-sacral crest in the sagittal plane (Figure 1).

#### 2.3.2. DIERS Formetric 4D Rasterstereographic System for Spinal Alignment

We used the DIERS Formetric 4D system (DIERS International GmbH, Schlangenbad, Germany) to assess spinal alignment once in each of the three sitting postures. The DIERS system projects raster lines onto the back surface and reconstructs a three-dimensional image of the trunk using stereophotogrammetry. The image acquisition time is approximately 0.1 s. Measurements were performed without markers or radiation exposure, and anatomical landmarks were automatically identified by the system. The parameters obtained included thoracic kyphosis angle, lumbar lordosis angle, trunk inclination, and lateral deviation.

The sternal angle represents the inclination of the sternum in the sagittal plane and reflects the anterior tilt of the upper chest, which is functionally related to thoracic posture and respiratory mechanics. Thoracic kyphosis describes the posterior convexity of the thoracic spine in the sagittal plane and serves as an indicator of overall postural alignment. The sacral angle corresponds to the sagittal inclination of the sacrum and determines pelvic orientation and lumbar curvature. Lumbar lordosis quantifies the anterior concavity of the lumbar spine. Trunk inclination represents the forward or backward tilt of the trunk, defined as the angle between the vertical axis and the line connecting the C7 vertebra to the midpoint of the sacrum. Lateral deviation quantifies the displacement of the spinal midline from the vertical axis, reflecting asymmetry in trunk posture (Figure 2).

### 2.4. Analysis

Categorical variables are expressed as absolute and relative frequencies, and quantitative variables as means and standard deviations. The Shapiro–Wilk test was used to assess the normality of data distribution. As some variables did not meet the assumptions of normality, nonparametric tests were applied.

Differences between the 2017 and 2024 cohorts were examined using the Mann–Whitney U test. Comparisons among the three postural conditions (passive, forced, and corrected sitting) within each cohort were performed using the Kruskal–Wallis test followed by Dunn’s post hoc test. Relationships between angular and alignment parameters were analyzed using Spearman’s rank correlation coefficients.

Effect sizes (Cohen’s *d*) and 95% confidence intervals were calculated to provide a measure of practical significance. Statistical analyses were conducted using IBM SPSS Statistics version 13.2 (IBM Corp., Armonk, NY, USA). The significance level was set at *p* < 0.05.

A post hoc power analysis performed using G*Power (v3.1.9.7) indicated that, for a medium effect size (Cohen’s *d* = 0.5), α = 0.05, and power (1 − β) = 0.95, the required sample size was 210 participants; therefore, the achieved sample of 356 ensured adequate statistical power.

## 3. Results

### 3.1. Spinal Alignment

No significant differences were observed between the 2017 and 2024 cohorts for any of the spinal alignment variables (all *p* > 0.05). Within each cohort, significant differences were found among the three sitting postures (*p* < 0.05) for the sternal angle, sacral angle, lumbar lordosis, and lateral deviation. Thoracic kyphosis and trunk inclination differed significantly between the passive sitting posture and the other postures (*p* < 0.05). These findings indicate that, regardless of the year of measurement, posture-specific variations were consistent and reproducible across both cohorts (Table 2).

### 3.2. Correlation Analysis by Posture

In the passive sitting posture, a moderate positive correlation was observed between the sternal angle and thoracic kyphosis (*r* = 0.667, 95% CI [0.61–0.72], *p* < 0.001) and between the sacral angle and lumbar lordosis (*r* = 0.679, 95% CI [0.62–0.73], *p* < 0.001). A moderate negative correlation was found between lumbar lordosis and trunk inclination (*r* = −0.407, 95% CI [−0.49–−0.32], *p* < 0.001).

In the forced sitting posture, moderate positive correlations were again noted between the sternal angle and thoracic kyphosis (*r* = 0.695, 95% CI [0.64–0.75], *p* < 0.001) and between the sacral angle and lumbar lordosis (*r* = 0.743, 95% CI [0.69–0.79], *p* < 0.001).

In the corrected posture, most correlations were weaker and not statistically significant, indicating reduced coupling between thoracic and pelvic parameters when an upright, corrected position was voluntarily adopted (Table 3).

## 4. Discussion

Body posture is shaped by both genetic and environmental factors. Environmental influences such as prolonged sitting, screen exposure, and reduced physical activity contribute to developmental variability and inter-individual differences in spinal alignment [20]. The present study compared standardized sitting spinal alignment in two independent cohorts of children assessed in 2017 and 2024 and analyzed correlations among sagittal alignment parameters in passive, forced, and corrected sitting postures.

Normal spinal curvature plays an essential role in distributing loads across passive spinal structures, maintaining pelvic alignment, and preventing discomfort or pain. Increased sitting time is associated with flattening of lumbar lordosis [21], and both thoracic kyphosis and lumbar lordosis have been linked to body composition indicators such as BMI, body fat percentage, and waist circumference [22,23,24]. Grabara et al. (2024) [23] emphasized that maintaining healthy body composition may help reduce the risk of spinal curvature abnormalities [23]. In the present study, no significant differences in spinal alignment parameters were observed between the 2017 and 2024 cohorts. This finding suggests that, under standardized laboratory conditions, sitting spinal alignment exhibits a degree of structural stability that is not easily altered by short-term environmental or behavioral changes.

Some previous reports have indicated an increased prevalence of postural abnormalities during the COVID-19 pandemic [12,25], which contrasts with the current findings. This discrepancy is likely due to methodological differences. Most earlier studies evaluated posture in the standing position and reported categorical outcomes such as the prevalence of “poor posture” or scoliosis, whereas the present research analyzed continuous angular parameters obtained in a controlled sitting posture [26]. Therefore, while population-level prevalence studies may sensitively capture minor deviations, standardized instrument-based measurements may reflect more stable morphological characteristics. It is also possible that lifestyle changes during the pandemic affected other health parameters-such as physical fitness, obesity, or muscle endurance-more strongly than static spinal geometry [23]. The absence of BMI differences between the cohorts supports this interpretation.

Optimal sitting posture has been described as involving anterior pelvic tilt, physiological lumbar lordosis, and an appropriate thoracic position [16]. In the current study, significant correlations were observed between the sternal angle and thoracic kyphosis and between the sacral angle and lumbar lordosis in both passive and forced sitting postures. These associations indicate coordinated segmental relationships within the sagittal spinal chain rather than isolated curvature changes. In the corrected posture, a sternal angle of approximately 65° was consistently associated with physiologically favorable thoracic kyphosis and lumbar lordosis. This value may therefore serve as a practical reference for posture assessment and education, rather than a prescriptive or “optimal” target. The sternal angle, as a simple external cue, could help children and educators monitor sitting posture without specialized instrumentation, supporting postural education in school and home settings.

### Limitations

Several limitations should be considered when interpreting these findings. First, the study design was cross-sectional and based on two independently recruited cohorts, which precludes causal inference about temporal or pandemic-related effects. Second, no behavioral or ergonomic data (e.g., physical activity, screen time, or workstation setup) were collected, limiting the ability to account for potential confounders. Third, the cohorts differed in demographic composition and were recruited from a single region in Poland, which may restrict the generalizability of results. Finally, the study did not include longitudinal follow-up to assess adaptive postural changes over time. Future research should incorporate lifestyle and environmental factors, prospective interventions, and diverse populations to validate the observed relationships and to confirm whether angle-based postural cues can effectively support preventive programs in children.

Moreover, the definition of an “ideal” or “optimal” sitting posture remains conceptually challenging and may vary depending on developmental, ergonomic, and cultural factors. Future studies should integrate behavioral monitoring, longitudinal designs, and interventional approaches to refine and validate angle-based postural assessment strategies [27,28,29,30,31].

## 5. Conclusions

In summary, this study compared standardized sitting spinal alignment in two independent cohorts of children assessed in 2017 and 2024. No significant between-cohort differences were found in any alignment parameter, suggesting overall stability of sagittal spinal geometry under controlled conditions. Within each cohort, posture-specific variations were observed, confirming systematic differences among passive, forced, and corrected sitting positions.

Strong correlations between the sternal angle and thoracic kyphosis and between the sacral angle and lumbar lordosis indicate consistent geometric relationships within the sagittal spinal chain. A sternal angle of approximately 65° was consistently associated with physiologically favorable alignment and may serve as a practical reference value for posture assessment and education in children.

## Figures and Tables

**Figure 1 children-12-01547-f001:**
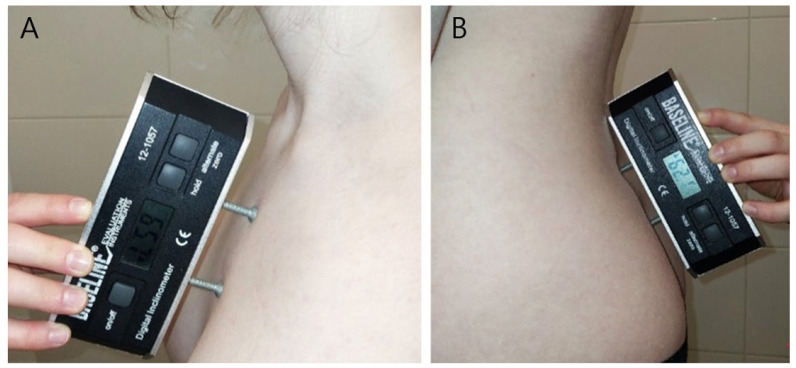
Measurement of the sternal angle (**A**) and sacral angle (**B**) using a Saunders digital inclinometer.

**Figure 2 children-12-01547-f002:**
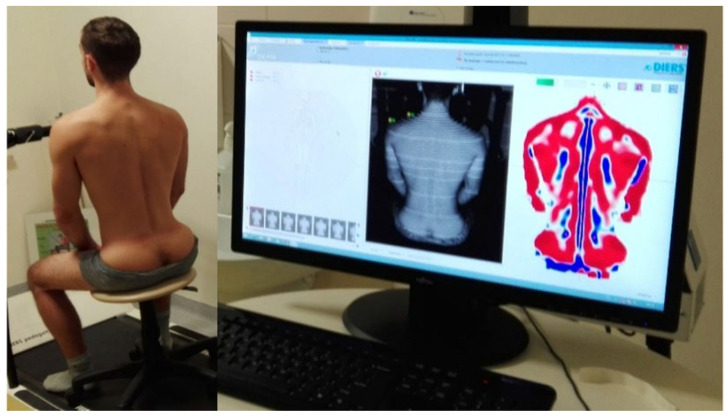
Assessment of spinal alignment parameters using the DIERS Formetric 4D system.

**Table 1 children-12-01547-t001:** General characteristics of participants (N = 356).

	2017 (*n* = 158)	2024 (*n* = 198)	*p*
Sex (male/female)	60 (37.97%)/98 (62.03%)	83 (41.92%)/115 (58.05%)	
Age	11.65 ± 1.19	11.59 ± 1.18	0.661
Weight	56.89 ± 8.45	56.44 ± 8.10	0.615
Height	164.63 ± 8.29	164.67 ±7.88	0.969
BMI	20.90 ± 2.06	20.75 ± 2.16	0.496

*p* < 0.05, *p* represents the difference between groups. BMI; Body Mass Index.

**Table 2 children-12-01547-t002:** Comparison of spinal alignment by posture before and after COVID-19.

Variables	Year	Passive ^a^	Forced ^b^	Corrected ^c^	*p* Value
Sternal angle (°)	2017	79.50 ± 8.69 ^a^	70.11 ± 11.16 ^b^	63.68 ± 2.15 ^c^	0.000 *
	2024	80.98 ± 8.90 ^a^	71.54 ± 10.97 ^b^	63.66 ± 2.19 ^c^	0.000 *
	*p* Value	0.117	0.226	0.913	
Sacral angle (°)	2017	83.62 ± 9.96 ^a^	91.58 ± 8.78 ^b^	113.28 ± 5.60 ^c^	0.000 *
	2024	82.00 ± 10.26 ^a^	90.95 ± 8.75 ^b^	112.45 ± 5.77 ^c^	0.000 *
	*p* Value	0.137	0.499	0.178	
Thoracic kyphosis (°)	2017	61.00 ± 12.25 ^a^	43.15 ± 13.65 ^b,c^	43.45 ± 3.30 ^b,c^	0.000 *
	2024	62.34 ± 12.62 ^a^	44.66 ± 13.49 ^b^	43.38 ± 3.22 ^c^	0.000 *
	*p* Value	0.318	0.298	0.912	
Lumbar lordosis (°)	2017	0.49 ± 15.96 ^a^	12.28 ± 12.45 ^b^	38.36 ± 4.34 ^c^	0.000 *
	2024	−1.12 ± 16.43 ^a^	12.46 ± 12.62 ^b^	38.02 ± 4.48 ^c^	0.000 *
	*p* Value	0.356	0.750	0.465	
Trunk inclination (°)	2017	14.67 ± 7.09 ^a^	2.22 ± 6.35 ^b,c^	3.36 ± 1.59 ^b,c^	0.000 *
	2024	15.30 ± 7.06 ^a^	2.69 ± 6.31 ^b,c^	3.40 ± 1.64 ^b,c^	0.000 *
	*p* Value	0.403	0.485	0.812	
Lateral deviation (mm)	2017	6.63 ± 3.62 ^a^	4.03 ± 1.88 ^b^	2.11 ± 1.27 ^c^	0.000 *
	2024	6.71 ± 3.55 ^a^	4.07 ± 1.93 ^b^	2.14 ± 1.26 ^c^	0.000 *
	*p* Value	0.825	0.824	0.854	

* *p* < 0.05. Values are presented as mean ± standard deviation. Same superscript alphabet means not statistically difference between groups based on nonparametric statistics Kruskal–Wallis and Dunn post hoc test.

**Table 3 children-12-01547-t003:** Comparison of correlations between spinal alignment according to posture.

	Variables	Sternal Angle	Sacral Angle	Thoracic Kyphosis	Lumbar Lordosis	Trunk Inclination	Lateral Deviation
Passive sitting	Sternal angle	1.000					
	Sacral angle	−0.257 *** (*p* = 0.000)	1.000				
	Thoracic kyphosis	0.667 *** (*p* = 0.000)	−0.213 *** (*p* = 0.000)	1.000			
	Lumbar lordosis	−0.171 ** (*p =* 0.001)	0.679 *** (*p* = 0.000)	−0.284 *** (*p* = 0.000)	1.000		
	Trunk inclination	0.218 *** (*p* = 0.000)	−0.296 *** (*p* = 0.000)	0.216 *** (*p* = 0.000)	−0.407 (*p* = 0.000)	1.000	
	Lateral deviation	0.212 *** (*p* =0.000)	−0.117 * (*p* = 0.028)	0.215 *** (*p* = 0.000)	−0.112 * (*p* = 0.035)	0.103 (*p* = 0.053)	1.000
Forced sitting	Sternal angle	1.000					
	Sacral angle	−0.280 *** (*p* = 0.000)	1.000				
	Thoracic kyphosis	0.695 *** (*p* = 0.000)	−0.270 *** (*p* = 0.000)	1.000			
	Lumbar lordosis	−0.145 ** (*p* = 0.006)	0.743 *** (*p* = 0.000)	−0.204 *** (*p* = 0.000)	1.000		
	Trunk inclination	0.202 *** (*p* = 0.000)	−0.269 *** (*p* = 0.000)	0.196 *** (*p* = 0.000)	−0.194 *** (*p* = 0.000)	1.000	
	Lateral deviation	−0.047 (*p* = 0.374)	0.111 * (*p* = 0.037)	−0.065 (*p* = 0.222)	0.169 ** (*p* = 0.001)	0.078 (*p* = 0.141)	1.000
Corrected sitting	Sternal angle	1.000					
	Sacral angle	0.078 (*p* = 0.143)	1.000				
	Thoracic kyphosis	0.119 * (*p* = 0.025)	0.082 (*p* = 0.121)	1.000			
	Lumbar lordosis	0.130 (*p* = 0.014)	0.308 *** (*p* = 0.000)	0.028 (*p* = 0.604)	1.000		
	Trunk inclination	−0.186 *** (*p* = 0.000)	−0.119 * (*p* = 0.025)	−0.166 ** (*p* = 0.002)	0.023 (*p* = 0.661)	1.000	
	Lateral deviation	−0.008 (*p* = 0.887)	0.080 (*p* = 0.135)	0.010 (*p* = 0.135)	0.057 (*p* = 0.284)	−0.014 (*p* = 0.794)	1.000

* *p* < 0.05, ** *p* < 0.01, *** *p* < 0.001.

## Data Availability

The datasets generated and analyzed during the current study are available from the corresponding author upon reasonable request. The data are not publicly available due to [restrictions related to personal data protection and the conditions of the ethics committee approval.].

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
