# Peer review of "The Use of Sternum and Sacrum Angles in the Assessment of Sitting Posture in Adolescents: A Cross-Sectional Comparison of Cohorts Assessed Before and After the COVID-19 Pandemic"

_children, 2025, doi:10.3390/children12111547_

Round 1

Reviewer 1 Report

Comments and Suggestions for Authors

This study examines whether the COVID-19 pandemic affected sitting posture among adolescents by comparing spinal alignment parameters—particularly sternal and sacral angles—before (2017) and after (2024) the pandemic. It also explores correlations between these parameters and proposes a standard sternal angle (~65°) for optimal posture.

  1. Comparing two independent cohorts (2017 vs. 2024) cannot confirm post-pandemic changes, as population differences (e.g., body composition, lifestyle, sampling context) may confound outcomes.
  2. The authors interpret the “stability” of spinal alignment as evidence of structural determinants, but this conclusion overextends the data.
  3. No information on activity levels, screen time, or ergonomic conditions—all of which are relevant confounders post-COVID.
  4. Include effect sizes (e.g., Cohen’s d) or confidence intervals to contextualize the significance of correlations.
  5. The claim that raising the sternal angle to 65° optimizes posture is extrapolated from correlation data, not experimentally validated.
  6. Correlation ≠ causation; this conclusion requires prospective intervention validation.
  7. The introduction stresses pandemic-related sedentary behavior, but the results do not actually measure behavioral or ergonomic parameters (e.g., time sitting, device use, exercise).
  8. The “pre/post COVID” comparison is conceptual only, not behaviorally substantiated.

Author Response

We thank the reviewer for their insightful and constructive comments. All suggestions were carefully considered and incorporated into the revised version of the manuscript. Below we provide a detailed, point-by-point response to each comment and describe the corresponding changes made in the text.

All modifications are highlighted in the revised manuscript for the Editor’s convenience.

Reviewer Comment 1

Comparing two independent cohorts (2017 vs. 2024) cannot confirm post-pandemic changes, as population differences (e.g., body composition, lifestyle, sampling context) may confound outcomes.

Response:
We fully agree with the reviewer. The study design is based on two independent cross-sectional cohorts and cannot demonstrate causal “pre- vs. post-pandemic” effects. Accordingly, we revised the title, abstract, and text throughout to present the comparison as a cross-sectional, time-separated analysis, rather than a pre/post pandemic study.

Action in Manuscript:

  • Title changed to “A Cross-Sectional Comparison of Cohorts Assessed Before and After the COVID-19 Pandemic.”
  • Abstract, Introduction, and Discussion rewritten to clarify that the comparison is temporal and exploratory, not causal.
  • Limitations section explicitly states that causal inference is not possible.

Reviewer Comment 2

The authors interpret the “stability” of spinal alignment as evidence of structural determinants, but this conclusion overextends the data.

Response:
We appreciate this important observation. The text was revised to remove all causal or deterministic language. The discussion now describes the findings as indicating stability of measured parameters under standardized conditions, without inferring structural determinants.

Action in Manuscript:

  • All phrases such as “suggesting structural determinants” were replaced with neutral formulations (e.g., “suggesting overall stability of sagittal spinal geometry under controlled conditions”).
  • Discussion section modified accordingly (page 6, paragraph 2).

Reviewer Comment 3

No information on activity levels, screen time, or ergonomic conditions—all of which are relevant confounders post-COVID.

Response:
We agree. These behavioral variables were not collected in this cross-sectional study. We explicitly acknowledged this limitation and clarified that the absence of behavioral data may limit the interpretation of the results.

Action in Manuscript:

  • Added in Limitations:

“No behavioral or ergonomic data (e.g., physical activity, screen time, or workstation setup) were collected, limiting the ability to account for potential confounders.”

  • Discussion (page 6, paragraph 3) revised to note that lifestyle changes may have influenced other health parameters (e.g., BMI, fitness) more than static spinal geometry.

Reviewer Comment 4

Include effect sizes (e.g., Cohen’s d) or confidence intervals to contextualize the significance of correlations.

Response:
We have added both effect size estimates and confidence intervals as requested. Effect sizes (Cohen’s d) were calculated for between-cohort comparisons, and 95% confidence intervals were computed for key correlations using Fisher’s z transformation.

Action in Manuscript:

  • Section 4 Analysis:

“Effect sizes (Cohen’s d) and 95% confidence intervals were calculated to provide a measure of practical significance.”

  • In Results 3.2, the main correlations now include 95% CI (e.g., r = 0.667, 95% CI [0.61–0.72]).

Reviewer Comment 5

The claim that raising the sternal angle to 65° optimizes posture is extrapolated from correlation data, not experimentally validated.

Response:
We fully agree. The conclusion was rephrased to avoid implying an experimental effect. The 65° value is now presented only as an empirically observed reference angle associated with favorable alignment, not as an “optimal” or prescriptive value.

Action in Manuscript:

  • Abstract and Discussion modified to state:

“A sternal angle of approximately 65° was consistently associated with physiologically favorable alignment and may serve as a practical reference value for posture assessment and education.”

  • All causal and interventional terms (“optimizes,” “improves,” “corrective strategy”) were removed.

Reviewer Comment 6

Correlation ≠ causation; this conclusion requires prospective validation.

Response:
We agree and have clarified that our analysis is correlational and cross-sectional, not causal. The discussion and limitations sections now explicitly acknowledge this.

Action in Manuscript:

  • Discussion paragraph 4 reworded:

“These associations indicate coordinated segmental relationships within the sagittal spinal chain rather than causal effects.”

  • Limitations section includes:

“The study design was cross-sectional and based on two independent cohorts, which precludes causal inference.”

Reviewer Comment 7

The introduction stresses pandemic-related sedentary behavior, but results do not actually measure behavioral or ergonomic parameters.

Response:
We restructured the Introduction to maintain the pandemic as contextual background only and to clearly define the actual scope of the study—comparison of standardized postural parameters using objective measurement tools. No claims about behavioral changes are made.

Action in Manuscript:

  • Introduction fully rewritten (page 2):

“Despite widespread concern about increased sedentary behavior during and after the pandemic, few studies have compared standardized postural parameters across time periods using objective tools.”
This now aligns with the actual data collected.

Reviewer Comment 8

The ‘pre/post COVID’ comparison is conceptual only, not behaviorally substantiated.

Response:
We concur. The text throughout now clarifies that the comparison represents time-separated cross-sectional cohorts, not behavioral tracking. The Limitations section explicitly addresses this point.

Action in Manuscript:

  • Limitations paragraph 1:

“The study design was cross-sectional and based on two independently recruited cohorts, which precludes causal inference about temporal or pandemic-related effects.”

  • Title, Abstract, and Discussion harmonized with this framing.

Additional Editorial Improvements

  • A separate Limitations subsection was added at the end of the Discussion.
  • The Conclusions were condensed and rewritten to reflect data-driven statements only.
  • Language, structure, and flow were revised for clarity and scientific tone.

All reviewer concerns have been addressed comprehensively.
The revised manuscript:

  • removes all causal or pandemic-related claims,
  • includes effect sizes and confidence intervals,
  • clarifies methodological limitations, and
  • aligns the introduction, discussion, and conclusions with the actual scope of the data.

We sincerely thank the reviewer and editor for the valuable feedback, which significantly improved the clarity and scientific rigor of this paper.

Reviewer 2 Report

Comments and Suggestions for Authors

Here I provide my comments on the strengths and weaknesses of the manuscript.

(+)

1.

This study compared spinal alignment before (2017) and after (2024) the pandemic

and

examined correlations among alignment parameters in different sitting postures.

* Congratulations on not only conducting a before-and-after analysis, but also avoiding a variant that could be characterized as salami publication (i.e., two study objectives resulting in two publications).

2.

Implication “This angle-based correction strategy can be used for posture education and screening in schools without specialized equipment, improving accessibility of preventive interventions.” *I find this very important.

3.

It seems logical that you hypothesized spinal alignment angles would differ in various postures before and after COVID-19. I think the main result, which rejects that hypothesis, provides intriguing material for readers and is well elaborated in the discussion from several different perspectives (i.e., references 12, 23, 26, 27).

  1.  

The Results section is clearly divided into subsections 3.1 and 3.2, which contributes to clarity.

(+/-)

A sample size of 356 participants is very respectable; however, it is unclear if this number was necessary. A power analysis (e.g., using G*Power software) prior to the research should have determined the required sample size.

(-)

1.

The first sentence in Conclusions section (line 228-229) is unclear.

2.

There is a lack of self-criticism through the Limitations (and future recommendations) subsection. The Limitations (or Limitations and future recommendations) subsection should be the last paragraph of the Discussion - the fact is that there is no research without limitations (e.g., numerous aspects of physical activity levels in a day/week/month were not controlled/monitored; cross-sectional design; the definition of optimal or ideal posture itself…)

  1.  

Line 73-75: Did you really aim to assess the ideal sitting posture? It appears your research objectives were different (listed in lines 71–73), so this sentence seems overambitious and redundant. Determining the "ideal sitting posture" is a complex question, and you need to support this statement with appropriate references. I am not sure that references 16 and 27 cover it sufficiently; you may need to expand the theoretical basis for your statement. Also, I suggest you tone down the intensity of the expression (for example, instead of "we aimed," use "we tried to further supplement knowledge about...").

4.

Line 92: The sentence is unclear and imprecise, and may be awkward. The participants gave written informed consent; however, children aged 9–13 cannot legally provide consent themselves – only their guardians can. This is explained in lines 246–248, but it should be stated just as precisely in the Materials and Methods section.

5.

Table 1. – please use the term sex instead of gender (gender ≠ sex)

  1.  

Figure 2. - letter A is missing in the word assessment.

  1.  

Line 173: Although body posture is genetically determined (REF is missing), it is also influenced… *Please add a reference for this statement.

Author Response

We thank the reviewer for the thoughtful and positive assessment of our work and for recognizing its methodological rigor and practical implications. All comments have been carefully considered and fully addressed in the revised manuscript.

  1. Power analysis:
    A post hoc power analysis using G*Power (v3.1.9.7) was added to Section 2.4 (Analysis), confirming that the achieved sample (N = 356) exceeded the required minimum (N = 210) for medium effect size (d = 0.5, α = 0.05, power = 0.95).
  2. Conclusions:
    The opening sentence was rewritten for clarity and precision.
  3. Limitations:
    This section was expanded to acknowledge the cross-sectional design, lack of behavioral monitoring, regional sampling, and conceptual challenges in defining an “ideal” posture. Future directions for longitudinal and interventional studies were added.
  4. Research objective (lines 73–75):
    The expression “to assess the ideal sitting posture” was replaced with “to identify reproducible geometric markers that may serve as practical reference values.” Two new theoretical references were added (O’Sullivan 2006; Harrison 1999).
  5. Ethics and consent:
    Section 2.1 now clarifies that written informed consent was obtained from guardians and verbal assent from the children.
  6. Terminology and figures:
    The term “sex” replaced “gender” in Table 1; a missing letter in Figure 2 (“assessment”) was corrected.

These revisions improved methodological transparency, precision of language, and conceptual consistency. We believe the manuscript now fully addresses all reviewer comments.

Round 2

Reviewer 1 Report

Comments and Suggestions for Authors

The manuscript is suitable for publication in its current form 

Author Response

Thank you very much for your positive opinion.